# Migration intentions and their impact on healthcare workers in a Lebanese public university hospital amid crises: A mixed-method study

Linda Abou-Abbas[1]*, Maya Hassan[2], Elizabeth Al Mounajed[1],
Mohamad Shafic Ramadan[1], Hala Dalal[1]

**1** International Committee of the Red Cross, Lebanon, **2** Faculty of Health Sciences (FHS), American University of Beirut, Lebanon

* labouabbas@icrc.org

## Abstract

### Background

The emigration of skilled healthcare workers (HCWs) seeking better opportunities poses major challenges to healthcare systems in low- and middle-income countries. Amidst ongoing economic and political crises, Lebanon is facing substantial healthcare workforce migration. This study explored the migration intentions of Lebanese HCWs, identifies key drivers, and proposes context-specific retention strategies.

### Methods

A mixed-methods design was employed at Rafik Hariri University Hospital (RHUH) in Beirut, Lebanon. A structured survey assessed migration intentions and associated drivers among frontline healthcare workers, while semi-structured interviews with department chiefs explored organizational and systemic factors influencing workforce retention. Quantitative data were analyzed to identify predictors of migration intentions, and qualitative data were thematically analyzed using a combined deductive–inductive approach.

### Results

Among 120 HCWs surveyed, 70% expressing intentions to migrate—primarily due to financial concerns (93%) and security issues (81%). Lower satisfaction with pay (adjusted OR= 0.85, 95% CI = 0.72–0.996) was significantly associated with migration intentions. Qualitative findings confirmed widespread staff migration since 2019, leading to critical shortages, heavier workloads, and department closures. Department chiefs emphasized the need for financial incentives, professional development opportunities, supportive management, and flexible scheduling to improve retention.

**Data availability statement:** The data supporting the findings of this study are owned by the International Committee of the Red Cross (ICRC) and are subject to legal and ethical restrictions related to participant confidentiality. Due to the contextual and sensitive nature of the data, full anonymization is not possible without removing key variables essential for replication and interpretation, including professional characteristics, and employment-related information. As a result, the dataset cannot be made publicly available. Access to the data may be granted to qualified researchers for replication or secondary analysis purposes, subject to approval by the ICRC and in accordance with its data protection and ethical policies. Data access requests should be directed to the ICRC through its Geneva Core mailbox at gva_core_mailbox@icrc.org.

**Funding:** This study was funded by the Agence Française de Développement (AFD) as part of its partnership project with the International Committee of the Red Cross (ICRC) and Rafik Hariri University Hospital (RHUH). AFD had no role in the conceptualization, design, data collection, analysis, decision to publish, or preparation of the manuscript.

**Competing interests:** The authors have declared that no competing interests exist.

**Abbreviations:** LMICs:low-and-middle-income countries; ICRC: International Committee of the Red Cross; HCW: Health care workers; UHC: Universal health coverage; RHUH: Rafik Hariri University Hospital; MJS: Measure of Job Satisfaction; AUB: American University of Beirut; SD: standard deviation

## Conclusions

HCW migration from Lebanon reflects a complex interplay of financial, professional, and systemic factors. Immediate and sustainable policy interventions—combining financial stabilization with workforce development and improved working conditions—are urgently required to preserve healthcare system functionality and resilience.

## Introduction

The concept of human capital flight, also known as "brain drain," describes "the emigration of highly skilled and educated individuals from one geographical area to another, typically from developing to developed countries" [1]. This migration is driven by professionals seeking better opportunities abroad, attracted by factors such as political stability, economic prospects, and improved living conditions. While the receiving country benefits from this influx of talent, often referred to as brain gain, the sending country faces significant challenges, particularly due to shortages of skilled workers [1].This phenomenon has become a major global concern, especially in the healthcare sector [1].

The continuous emigration of healthcare workers (HCW) from low- and middle-income countries exacerbates economic disparities and creates critical shortages within healthcare systems [1]. Consequently, many of these countries struggle to maintain the minimum ratio of HCW required for effective health coverage [2], creating challenges in providing essential medical services to their populations. Moreover, mass migration raises fundamental questions about the long-term sustainability and quality of healthcare services in these countries [3,4].

In the destination country, migrant HCW play a pivotal role in bolstering the response to health emergencies and in the pursuit of universal health coverage (UHC). Their skills, expertise, and dedication help fill critical workforce gaps, particularly in underserved areas or communities with limited access to medical services. In times of crisis, such as pandemics or natural disasters, these professionals are often at the forefront, providing critical care and support to affected populations. Moreover, in the context of achieving UHC, migrant health workers contribute significantly to filling gaps in healthcare services, especially in underserved regions or communities with limited access to medical facilities [5].

HCW migration is shaped by a complex interplay of push and pull factors at both the individual and systemic levels [6–10]. On an individual scale, factors such as better job opportunities, improved working conditions, higher salaries, and enhanced career prospects act as strong motivators for migration [9,11]. Simultaneously, systemic issues such as under-resourced and poorly organized healthcare systems, limited professional development opportunities, and political instability—characterized by recurrent government paralysis, policy uncertainty, and weak institutional governance—can force healthcare workers to seek employment abroad [6,7,9]. Moreover, macro-level factors such as economic globalization and international trade agreements further influence the global distribution of HCW, potentially worsening

workforce migration from LMICs [9,12,13]. Understanding these factors is essential for developing effective policies aimed at retaining HCW in their home countries [14].

Lebanon, situated in the Middle East and North Africa (MENA) region, faces significant challenges in its healthcare sector that are exacerbated by enduring crises. Historically, Lebanon has experienced significant brain drain among HCW, with earlier waves of migration linked to the civil war (1975–1990) and its aftermath, followed by the partial return of professionals during the post-war reconstruction period [15–17]. Despite having a nurse density slightly below the global average [18] and higher medical doctor density [19], research over the past two decades has high-lighted push and pull factors influencing HCW migration, including societal pressures favoring international specialty training and a prevailing culture of migration among medical graduates [20]. Specifically, a 2007 study revealed that challenging work conditions and limited professional autonomy drove one in five nurses to migrate, primarily to Gulf countries [21]. A more recent study conducted between 2017 and 2018 further explored the migration patterns of Lebanese nurses, identifying unsatisfactory salary and benefits, better opportunities abroad, and limited career advancement as primary reasons for leaving. Despite these challenges, many nurses expressed a willingness to return if they offered improved incentives such as financial benefits, professional development, and career advance-ment opportunities [22].

Since late 2019, Lebanon has experienced a series of overlapping crises, beginning with a severe economic collapse, followed by the COVID-19 pandemic in 2020–2021, and the Beirut port explosion in August 2020, which collectively placed unprecedented strain on the country's social and healthcare systems [23]. As a result, the purchasing power of Lebanese citizens has plummeted, making it increasingly difficult for individuals and families to cover basic needs. The devaluation of the Lebanese pound has further eroded the value of salaries, significantly reducing their real value in terms of purchasing goods and services. This economic downturn has created a challenging environment where many are strug-gling to make ends meet despite being employed.

Lebanon's healthcare system has been deeply affected by the ongoing economic crisis, resulting in severe shortages of medications and medical supplies and growing frustration among both providers and patients. Governmental hospi-tals—29 out of the country's 157 hospitals—have been particularly strained by inadequate funding, deteriorating infra-structure, and the emigration of 30–40% of healthcare professionals seeking better opportunities abroad [24]. Studies in Lebanon have highlighted that physician emigration is largely driven by declining income, limited career advancement, reduced quality of care, unhealthy work environments, and the broader political and socio-economic instability. In contrast, factors such as emotional attachment to the profession and workplace recognition can help retain staff [25]. Collectively, these challenges underscore the urgent need for strategies to stabilize and strengthen the healthcare workforce. Public hospitals are especially affected, facing delayed or reduced salaries, heavier workloads due to staff shortages, limited patient care resources, and heightened occupational stress—all exacerbated by persistent funding deficits and opera-tional constraints.

This study was conducted at Rafik Hariri University Hospital (RHUH), the largest public university and teaching hospital in Beirut, the capital of Lebanon, which has an estimated population of approximately 2.4 million residents in its metropolitan area. As a key institution in the Lebanese healthcare system, RHUH provides high-quality medical services to Lebanese citizens and patients from all regions, including specialized care in trauma, cardiac, and rare conditions, and has played a central role in educating healthcare professionals. Although the hospital was initially equipped with 430 beds, 14 operating rooms, and 50 suite hotel accommodations, its capacity is currently only partially operational due to the ongoing crisis, with the burn unit closed and hotel accommodations unavailable [26]. These operational challenges create a demanding environment for HCW, reflecting the broader national context of workforce instability and resource constraints. By situating the study within this setting, it captures the experiences and migration intentions of HCW across multiple professional categories within a major public institution, underscoring the relevance and contribution of this research.

In this context, the current study used a mixed-methods approach to assess migration intentions of HCWs at RHUH. The quantitative component surveyed frontline HCWs to identify their intentions and key migration drivers, while the qualitative component explored management-level perspectives to understand systemic and organizational factors. Integrating both viewpoints allows identification of actionable strategies to mitigate workforce migration and support Lebanon's healthcare system.

## Materials and methods

### Study design and setting

This study employed a mixed-methods approach, integrating quantitative surveys of HCW and qualitative key informant interviews with hospital department chiefs. The study was conducted at RHUH, the largest public university hospital in Beirut, Lebanon.

### Study participants

**Quantitative phase.** HCW employed at RHUH, including doctors, nurses, pharmacists, dietitians, inhalation specialists, laboratory technicians, phlebotomists, physiotherapists, clerks, and secretaries. Medical residents and department chiefs were excluded because residents hold temporary training positions with different career trajectories, and department chiefs occupy managerial roles; including them in the quantitative survey could bias estimates of migration intentions among the regular workforces.

**Qualitative phase.** Key informant interviews were conducted with department chiefs and other managerial staff.

- **Inclusion:** Supervisors, managers, and heads of departments.

- **Exclusion:** All other hospital staff who participated in the quantitative survey, including attending doctors, nurses, pharmacists, dietitians, inhalation specialists, laboratory technicians, phlebotomists, physiotherapists, clerks, and secretaries.

This distinction allowed a bottom-up approach through the quantitative survey capturing the intentions and perspectives of HCWs, and a top-down approach through qualitative interviews capturing managerial insights on systemic drivers of workforce migration.

### Ethical considerations

Ethical approval for the study was obtained from the Institutional Review Board at RHUH (10 Feb 2022) and the ethical review board of the International Committee of the Red Cross (ICRC) (reference number DP_CORE 22/00004-CGB/bap). All participants were fully informed about the purpose, procedures, risks, and benefits of the study before providing consent.

- **Quantitative Phase:** Consent was obtained via a yes/no question: "Did you read and agree to fill out the questionnaire?" Participants who answered "No" were not allowed to proceed, and the survey ended.

- **Qualitative Phase:** Written informed consent was obtained from all key informants prior to the interviews.

Participation was entirely voluntary, without coercion or undue influence, and participants had the right to withdraw from the study at any time without penalty. All collected data were kept strictly confidential and anonymized to protect participants' privacy. Access to the collected data was restricted to the research team and was utilized solely for research purposes. The data were stored on password-protected computers in a secure location, will be retained for five years, and then securely destroyed in accordance with institutional and International Committee of the Red Cross (ICRC) data management policies.

## Sampling strategy

The target population for the quantitative phase consisted of approximately 579 HCW employed at RHUH, including doctors, nurses, pharmacists, dietitians, inhalation specialists, laboratory technicians, phlebotomists, physiotherapists, clerks, and secretaries. This number represents the total eligible staff at RHUH at the time of the study, as obtained from the hospital's human resources and administrative registries. A stratified sampling approach was used to ensure proportional representation across various professional categories. To ensure proportional representation, a stratified sampling approach was used based on staff distribution: nurses (n = 400), attending doctors (n = 100), laboratory technicians and phlebotomists (n = 30), Administrative and support staff (n = 20), pharmacists (n = 12), inhalation specialists (n = 10), physiotherapists (n = 5), dietitians (n = 2). This approach ensured that the perspectives of all occupational groups were adequately represented in the study. The survey was then distributed to all eligible staff within each stratum.

For the qualitative component, purposive sampling was employed to select departmental chiefs. Data collection and analysis were conducted concurrently. Data saturation was achieved when no new information or insights emerged from subsequent interviews, ensuring a thorough exploration of the topic and the validity and robustness of the findings [27]. After twelve interviews, no new themes emerged, confirming that data saturation had been reached.

## Data collection tools

The data collection phase took place from 30 April 2022 to 15 July 2022. The research team developed a self-administered questionnaire following an exhaustive review of numerous pertinent studies conducted within the region [18,28,29]. The questionnaire underwent expert panel review, translation into Arabic, and content validation across Arabic and English through back-translation. It consists of four main sections:

- The sociodemographic characteristics included age group, gender, marital status, whether they had children, and educational level.

- Work experience included role or position held within the hospital, department affiliation, employment status (whether permanent or on contract), cumulative years of professional experience, duration of tenure specifically within the RHUH, average weekly working hours, monthly salary, and any concurrent employment in healthcare facilities external to the RHUH.

- Job Satisfaction: The Measure of Job Satisfaction (MJS) scale, originally developed by Traynor and Wade (1993) [30], has been previously utilized in Lebanon to assess the job satisfaction of healthcare providers [22]. For the purposes of this study at RHUH, the MJS was tailored to suit the hospital's context. This scale encompasses 33 items with six distinct subscales: personal satisfaction (6 items), satisfaction with workload (6 items), satisfaction with professional support (8 items), training (5 items), satisfaction with salary (3 items), and satisfaction with prospects (5 items). Participants were asked to rate their satisfaction levels on each subscale using a 5-point Likert scale ranging from 1 (very dissatisfied) to 5 (very satisfied). For scoring, each subscale score was calculated by summing the items within the subscale and transforming it to a 0–5 scale. Higher scores indicate greater satisfaction. Through internal consistency analysis, the MJS has demonstrated high Cronbach's alpha values for both individual subscales and the overall scale, ranging from 0.85 to 0.95, confirming the instrument's reliability.

- HCW migration intentions were assessed using the question: "Are you planning to migrate outside of Lebanon in the future?" with response options of 1 = "Yes" and 2 = "No." Respondents who answered "Yes" were subsequently asked to complete structured follow-up questions assessing their primary reasons for migration, using predefined response options (e.g., financial concerns, security issues, professional development, and systemic factors). In addition, all participants were asked one open-ended question inviting them to describe potential strategies or conditions that might motivate them to remain working at the hospital.

For the qualitative part, key informant interviews were conducted to complement the quantitative data collection. An interview topic guide (S1 Appendix) was developed for participants in managerial positions to collect detailed information on workforce migration. The topic guide included questions grouped under five main categories:

1. General Idea of HCW' Migration

2. Reasons for Migration

3. Impact of Migration

4. Current Retention Strategies

5. Future Retention Strategies

6. Additional concerns and questions

### Data collection procedure

First, HCW were sent an invitation along with a consent form via WhatsApp groups to complete an online questionnaire on Microsoft Office surveys. The reason for choosing this data collection method was the COVID-19 situation, which prevented the research team from conducting structured interviews with the participants. The questionnaire was available in both English and Arabic. Three reminders were sent to the 550 eligible participants, each one week apart. All participants provided consent before completing the survey. The questionnaire was designed to be concise, requiring approximately 10 minutes to complete, ensuring that it was manageable for HCW to fit into their busy schedules.

Following the qualitative phase of the study, 12 key informant interviews were conducted between 30 June 2022 and 15 July 2022. The interviews were primarily conducted in person (10 interviews), with an additional two conducted via telephone. A female MD student from the American University of Beirut (AUB), currently pursuing her master's degree, conducted the interviews. She has received rigorous training in qualitative research methods through her master's program at AUB.

Participants were interviewed in their private offices or in a private meeting room at the hospital, ensuring a confidential environment. All interviews were recorded for accuracy using a digital voice recorder after obtaining consent.

Prior to study commencement, efforts were made to establish transparency and manage potential biases related to the researcher-participant relationship. Participants were fully informed about the interviewer's background and qualifications as well as her role as a student researcher conducting this study as part of the academic requirements under the ICRC's supervision. The interviews lasted for approximately one hour.

During the study, new information and perspectives on HCW migration became increasingly scarce, while recurring themes and insights emerged across participants. This consistent repetition indicated that data saturation had been reached, as further data collection was unlikely to yield additional insights.

### Data analysis

For the quantitative analysis, IBM SPSS Statistics 27.0 software was utilized. Descriptive statistics were presented as means and standard deviations (SDs) for continuous variables and as frequencies and proportions for categorical variables. Bivariate analyses using chi-square tests and simple logistic regressions were first conducted to explore crude associations between migration intentions and participants' sociodemographic characteristics and job satisfaction factors. Variables showing an association at a significance level of $p \leq 0.20$ in the bivariate analysis, in addition to those deemed theoretically relevant based on prior literature, were included in the multivariate logistic regression model. This two-step approach allowed the identification of independent predictors of migration intention while adjusting for potential confounders.

The multivariate model was built using the enter method, and model fit was assessed with the Hosmer–Lemeshow test. In all analyses, a two-tailed p value < 0.05 was considered statistically significant.

For the qualitative analysis, interviews were conducted in Arabic, transcribed verbatim, and translated into English by a member of the research team. Translations were reviewed by another team member for accuracy and consistency with the original transcripts. Any discrepancies were resolved through discussion within the research team. The transcripts were then imported into Dedoose software for analysis. A combined deductive–inductive approach was used, and thematic analysis followed Braun and Clarke's (2006) six-step process: (1) familiarization with the data, (2) generating initial codes, (3) searching for themes, (4) reviewing themes, (5) defining and naming themes, and (6) producing the final report [31].

The initial coding framework was developed deductively, informed by the topic guide, existing literature, and conceptual frameworks on healthcare worker migration—specifically, Hajian et al.'s [9] macro–meso–micro categorization and the "push–pull" and "stick–stay" models described by Oberoi and Lin [10]. As coding progressed, new codes and subthemes that emerged from participants' narratives were incorporated inductively. The framework was refined iteratively through team discussions until agreement was reached on the final coding tree, which is illustrated in Fig 1.The qualitative methods and reporting of results adhered to the Consolidated Criteria for Reporting Qualitative Studies (COREQ) guidelines [32].

## Results

### Study participants

The quantitative survey targeted all eligible staff at RHUH (N = 550), of whom 120 HCWs completed the online survey, resulting in a response rate of 21.8%. Reasons for non-participation were not systematically collected. For the qualitative phase, 12 key informants (department chiefs and managerial staff) were purposively selected, separate from the quantitative participants, to capture managerial perspectives on workforce migration and retention strategies. Fig 2 presents the flowchart of participant selection for the quantitative and qualitative phases of the study.

The demographic characteristics of the study participants are presented in Table 1. Of the total respondents, 65.0% were females (n = 78), and 50.8% (n = 42) were aged between 35 and 44 years. Approximately half of the respondents were married (54.2%) and had children (55.0%). Educational backgrounds varied, with 45.0% holding a Baccalaureate or license technique degree, 31.7% a bachelor's degree, and 23.3% a Master's/PhD. More than half of the participants (58.3%) earned between $65 and $129, while only 5.0% earned more than $260.

Of the total participants, 60.8% were nurses. More than half of the respondents were contractor employees (59.2%), whereas 40.8% were permanent employees. With respect to years of work experience, approximately 62% of surveyed HCW had more than 10 years of experience in health institutions, and 61.7% had 11–20 years of experience in RHUH. Of the total respondents, 20.8% had another job. Approximately one-quarter of the participants (27.5%) worked more than 40 hours per week (Table 2).

### Intentions and drivers of HCW migration

Of the total respondents, 70.0% expressed their intention to migrate. The primary reasons for their migration intentions included financial issues such as unsatisfactory benefits or salaries (93.0%) and lack of job stability (30.0%). The work environment also played a significant role, with poor work conditions (55.0%), lack of a supportive environment (50.0%), and heavy workload (33.0%) being major concerns. Security issues were highlighted by 81.0% of respondents, while 37.0% cited a lack of professional development opportunities. Additionally, some respondents reported other personal reasons, such as the need to move with their families (17.0%) and searching for opportunities outside the hospital sector (15.0%) (Fig 3).

Bivariate analyses examining the association between sociodemographic characteristics and the likelihood of migrating outside Lebanon revealed that HCW earning a salary between $65 and $129 were more likely to express intentions to migrate compared to those earning less than $65 (P-value = 0.027) (Table 1). No statistically significant associations were

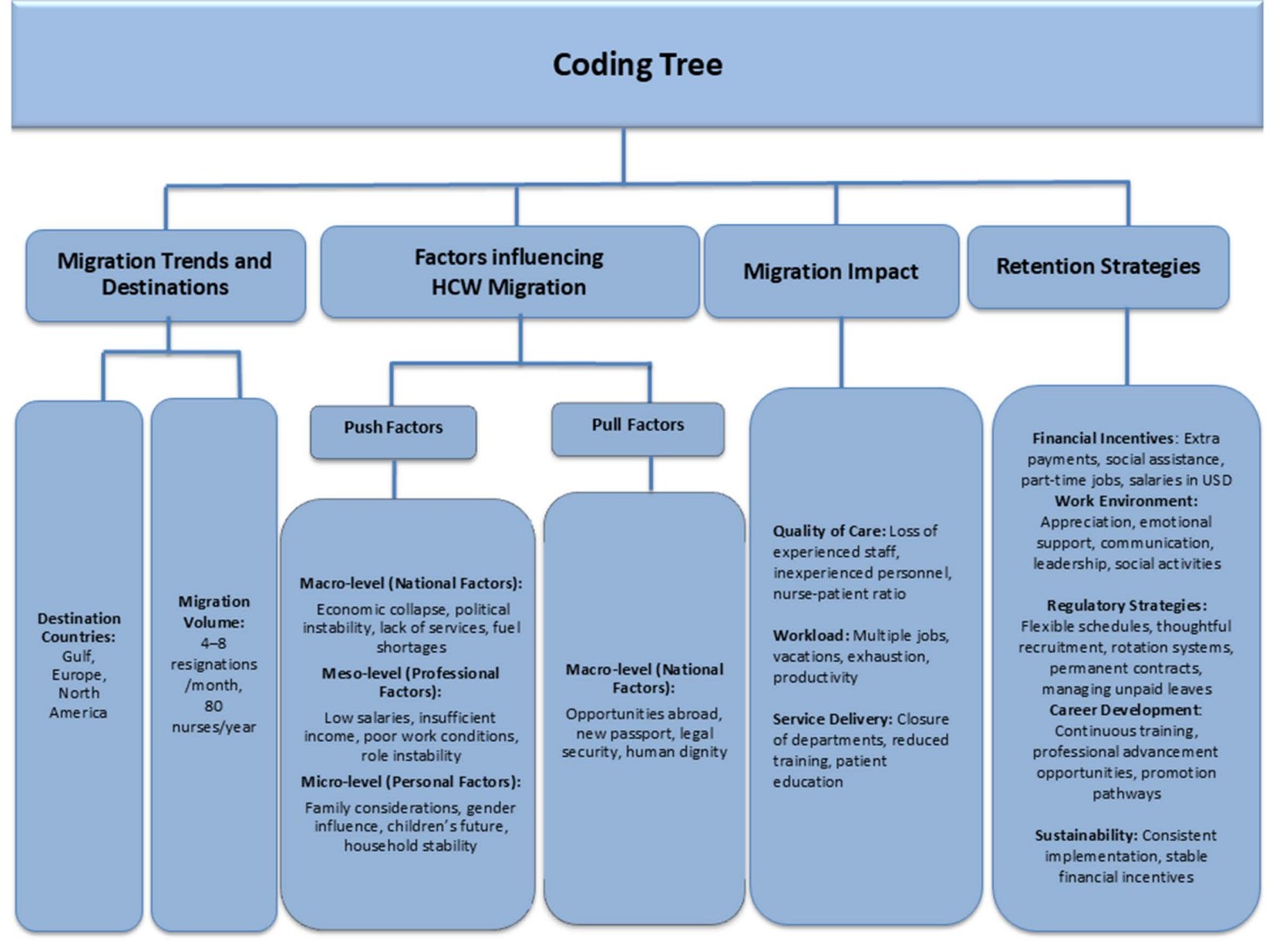

**Fig 1. Coding tree.**

found between migration intentions and the professional characteristics of HCW, as shown in Table 2. Additionally, our analysis revealed statistically significant associations between migration intention and personal satisfaction (OR= 0.35, 95% CI = 0.18–0.70), satisfaction with training (OR=0.46, 95% CI = 0.29–0.75), pay (OR=0.48, 95% 0.31–0.73), and prospects (OR= 0.47, 95% CI = 0.29–0.75) (Table 3). The multiple logistic regression analysis showed that higher satisfaction with pay was associated with lower odds of migration intention (adjusted OR = 0.82, 95% CI = 0.71–0.96).

## Staff suggestions for improving retention at RHUH

Participants were asked an open-ended question regarding measures that could encourage them to remain at RHUH. The predominant themes were financial compensation and support, such as paying part of the salary in dollars, increasing salaries to match inflation, and offering bonuses and incentives. They also emphasized professional growth and development, highlighting the need for training opportunities, clear pathways for promotion, and recognition of qualifications

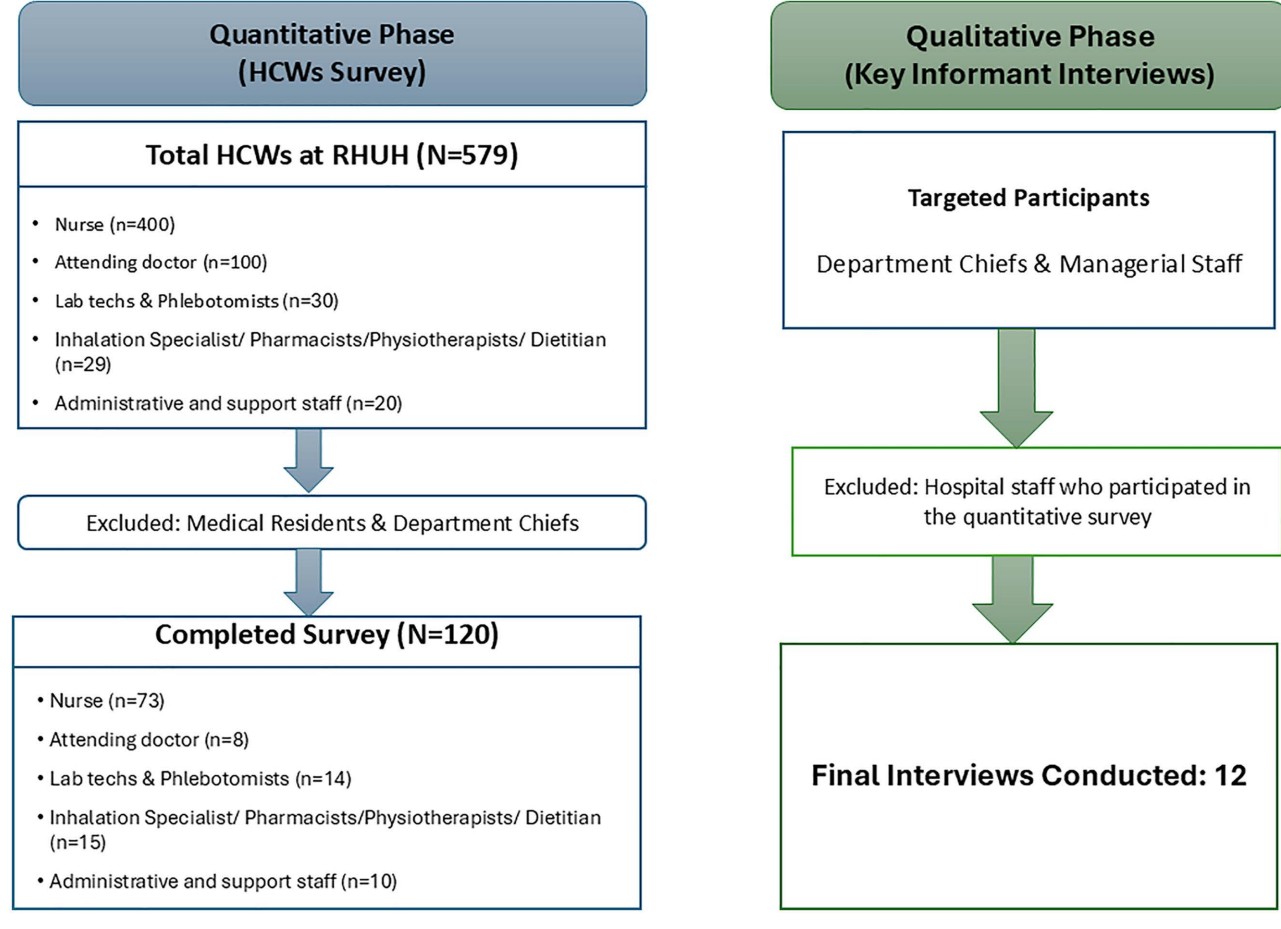

**Fig 2. Flowchart of participant selection and study design for the mixed-methods study at RHUH.**

and experience. Additionally, employees stressed the importance of improving the work environment by ensuring adequate resources, providing a supportive atmosphere, and reducing workloads. Benefits such as comprehensive health insurance, educational support for children, and retirement plans were also mentioned. Finally, the need for better management practices, including fair treatment, transparent decision-making, and effective communication, was reported (S1 Table).

**Findings from key informant interviews:**

A total of twelve semi-structured interviews were conducted with RHUH department chiefs. In total, 12 participants (8 females and 4 males) took part in the study, with 5–20 years of professional experience in total.

Themes and subthemes identified in the analysis are summarized in the coding framework (Fig 1) and elaborated below with supporting quotations.

1. Migration trends and destinations

Key informants emphasized the widespread migration of HCW since 2019 due to Lebanon's economic crisis. Most HCW are migrating or planning to migrate to Gulf countries (Iraq, United Arab of Emirates (UAE), Qatar, Saudi Arabia), followed by European destinations (France, Belgium, United Kingdom), and some to Canada and the United States of America (USA). Although precise numbers are unavailable, estimates suggest 4–8 job resignations per month on average, with

**Table 1. Sociodemographic characteristics of the study participants and Bivariate analysis showing the associations with migration intentions among HCWs (N = 120).**

| | | Migration Intention | | |
|---|---|---|---|---|
| | All (N = 120) | No (n = 36) | Yes (n = 84) | P-value |
| **Position n (%)** | | | | 0.174 |
| Nurses | 73 (60.8) | 20(27.4) | 53(72.6) | |
| Laboratory Staff/Phlebotomists | 14 (11.7) | 2(14.3) | 12(85.7) | |
| Administrative and support staff | 10 (8.3) | 6(60.0) | 4(40.0) | |
| Doctors | 8 (6.7) | 3(37.5) | 5(62.5) | |
| Others* | 15(12.5) | 5(33.3) | 10(66.7) | |
| **Employment Status n (%)** | | | | 0.841 |
| Contract | 71 (59.2) | 22(31) | 49(69) | |
| Permanent | 49 (40.8) | 14(28.6) | 35(71.4) | |
| **Years of work experience n (%)** | | | | 0.634 |
| <5 | 26 (21.7) | 7(26.9) | 19(73.1) | |
| 5–10 | 20 (16.7) | 8(40.0) | 12(60.0) | |
| >10 | 74 (61.7) | 21(28.4) | 53(71.6) | |
| **Years of experience in RHUH n (%)** | | | | 0.443 |
| <5 | 33 (27.5) | 9(27.3) | 24(72.7) | |
| 5–10 | 13 (10.8) | 6(46.2) | 7(53.8) | |
| 11–20 | 74 (61.7) | 21(28.4) | 53(71.6) | |
| **Work hours n (%)** | | | | 0.505 |
| <40 hours/week | 87 (72.5) | 28(32.2) | 59(67.8) | |
| ≥ 40 hours/week | 33 (27.5) | 8(24.2) | 25(75.8) | |
| **Working in another institution n (%)** | | | | 0.139 |
| No | 95 (79.2) | 32(33.7) | 63(66.3) | |
| Yes | 25(20.8) | 4(16.0) | 21(84.0) | |

N, n frequency, % percentage, * include Inhalation Specialist/ Pharmacists/Physiotherapists & Dietitian. P-value less than 0.05 were considered significant.

one participant noting 80 nurses migrating within a year, underscoring the substantial impact of migration on healthcare staffing in Lebanon.

*"In one year, 80 nurses left… this is not an exaggeration... 80 nurses can open a hospital." – Key informant 6*

2. Factors influencing HCW migration

Analysis of the key informant interviews revealed that HCW migration decisions are influenced by a combination of push and pull factors operating across macro, meso, and micro levels. This thematic organization follows the Hajian framework and highlights structural, professional, and personal influences.

a. Push factors:

First, at the macro level, participants repeatedly highlighted how the country's economic collapse, political instability, and lack of basic services are driving HCW to migrate. Many highlighted that the collapse of the Lebanese pound and the rising cost of living had created unprecedented financial strain, forcing them to consider leaving the country for the first time in decades.

**Table 2. Professional characteristics of the study respondents and Bivariate analysis of the associations with migration intention.**

| Variable | All (N = 120) | Migration Intention No (n = 36) | Yes (n = 84) | P-value |
|---|---|---|---|---|
| **Gender n (%)** | | | | 0.149 |
| Male | 42 (35) | 9(21.4) | 33(78.6) | |
| Female | 78 (65) | 27(34.6) | 51(65.4) | |
| **Age n (%)** | | | | 0.977 |
| <35 years | 45 (37.5) | 14(31.1) | 31(68.9) | |
| 35–44 years | 61 (50.8) | 18(29.5) | 43(70.5) | |
| ≥45 years | 14 (11.7) | 4(28.6) | 10(71.4) | |
| **Marital Status n (%)** | | | | 0.690 |
| Unmarried | 55 (45.8) | 15(27.3) | 40(72.7) | |
| Married | 65 (54.2) | 21(32.3) | 44(67.7) | |
| **Having Children n (%)** | | | | 0,692 |
| No | 54 (45.0) | 15(27.8) | 39(72.2) | |
| Yes | 66 (55.0) | 21(31.8) | 45(68.2) | |
| **Number of Children n (%)** | | | | 0.763 |
| 0 | 54 (45) | 15(27.8) | 39(72.2) | |
| 1-2 | 38 (31.7) | 11(28.9) | 27(71.1) | |
| ≥3 | 28 (23.3) | 10(35.7) | 18(64.3) | |
| **Educational Background n (%)** | | | | 0.256 |
| Baccalaureate/License Technique | 54 (45) | 20(37) | 34(63.0) | |
| Bachelor | 38 (31.7) | 8(21.1) | 30(78.9) | |
| Master's/PhD | 28 (23.3) | 8(28.6) | 20(71.4) | |
| **Monthly Salary Range n (%)** | | | | **0.027** |
| < 65$ | 19 (15.8) | 9(47.4) | 10(52.6) | |
| 65–129$ | 70 (58.3) | 15(21.4) | 55(78.6) | |
| >130 | 31 (25.8) | 12(38.7) | 19(61.3) | |

†Includes pharmacist, dietitian, physiotherapist, inhalation specialists, N frequency, % percentage, P value less than 0.05 is considered significant

"We are living the economic crisis… the dollar-Lebanese pound devaluation… the living situation: electricity cut… the high cost of living… I graduated from AUB 20 years ago… this is the first time since 20 years that I started thinking of migration…" – Key informant 5.

In addition to economic hardship, several key informants described how frequent political conflicts, and insecurity created an unstable environment that fueled uncertainty about the future.

"We, as Lebanese, have a fear of things that we should not be afraid of… I discovered that I feel insecure here…" – Key informant 7.

Some participants reported that the shortages in essential services, such as fuel, further burdened their daily lives and reinforced the need to leave the country.

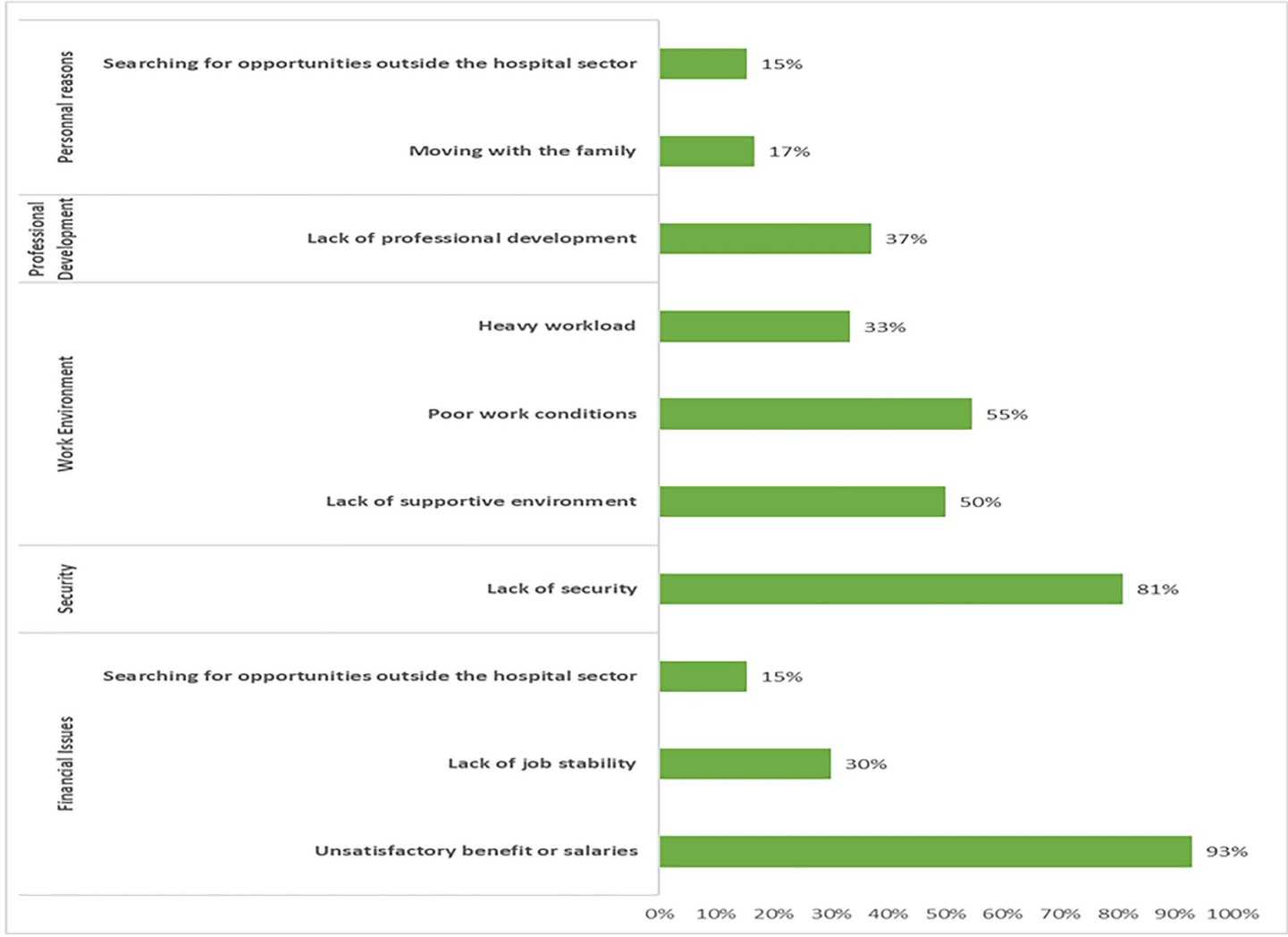

**Fig 3. Reasons for HCW migration intentions at RHUH (N = 120).**

**Table 3. Mean satisfaction scores on the job satisfaction scale and logistic regression analyses of the associations between job satisfaction and migration intentions among HCW (N = 120).**

| | All (N=120) | Migration Intention | | Simple Logistic Regression | Multiple Logistic Regression |
|---|---|---|---|---|---|
| | Mean (SD) | No (n=36) | Yes (n=84) | OR (95%CI) | Adjusted OR (95%CI) |
| **Subscales** | | | | | |
| Personal satisfaction | 3.3(0.6) | 3.5(0.6) | 3.2(0.6) | **0.35(0.18-0.70)** | |
| Satisfaction with workload | 3.2(0.6) | 3.4(0.7) | 3.1(0.6) | 0.54(0.28-1.03) | |
| Satisfaction with Professional Support | 3.3(0.8) | 3.4(0.6) | 3.2(0.8) | 0.69(0.40-1.17) | |
| Satisfaction with Training | 2.5(0.9) | 3.0(0.9) | 2.4(0.8) | **0.46(0.29-0.75)** | |
| Satisfaction with Pay* | 1.0(1.0) | 2.0(2.0) | 1.0(0.7) | **0.48(0.31-0.73)** | **0.82(0.71-0.96)** |
| Satisfaction with Prospects | 2.6(0.9) | 3.1(0.8) | 2.4(0.9) | **0.47(0.29-0.75)** | |

OR Odds ratios, CI confidence intervals, * medians and interquartile ranges are reported, and a P value less than 0.05 was considered to indicate statistical significance.

Second, at the meso level, income and work conditions were identified as key determinants of migration. Most key informants (8 out of 12) indicated that their current salaries are insufficient to cover basic living expenses due to the economic situation and the dollar-Lebanese pound devaluation. One participant explained,

*"The dollar rate is increasing... and we are still getting our salary on the 1,500 rates... the income is less than the consumption" -Key Informant 1, referring to salaries being calculated based on the old official exchange rate of 1,500 LBP per US dollar, which has not adjusted to rising living costs.*

While another added, *"The income is not enough to live in this country... this is a direct reason that is forcing staff to migrate"-Key Informant 3.*

*"One of the staff traveled for a salary of $700… imagine that! We were very surprised by that salary… he said that he is sending his family $300 to live…" – Key Informant 10*

In addition to financial pressures, work conditions such as frequent changes in job assignments were cited as influential factors. As one participant noted, *"The job keeps changing, and it's hard to feel motivated when there is no stability in your role" – Key Informant 4.*

Finally, at the micro level, personal and family considerations influenced HCW decisions to migrate. One key informant noted that gender played a role, with female staff often more dependent on their families' decisions regarding whether to stay or move abroad. Beyond gender, participants emphasized the importance of securing a better future for their children and maintaining household stability. As one participant explained,

*"One of us is traveling to secure a future for his/her children… some people are not affording to buy toys for their children… now, you go to the supermarket and check which brand is cheaper… we did not use to look at prices…" – Key Informant 9.*

b. Pull factors in other countries

At the same time, participants described macro-level opportunities abroad that made migration appealing. One participant noted,

*"Living in a country where people are treated with respect and human rights are protected… this is very important for me and my family" – Key Informant 2.*

Although the interview guide did not specifically explore factors attracting HCW to migrate, a few participants spontaneously mentioned certain pull factors in other countries. Two participants highlighted the opportunity to obtain another passport, which offers greater mobility and legal security. In addition, living in a country where human dignity is respected and valued was reported by two participants as an appealing factor. While these findings were mentioned by only a small number of participants, they suggest that structural and societal conditions in destination countries may also influence HCW' intentions to migrate.

3. Migration impact

Eight key informants mentioned that the loss of qualified and experienced HCW is a substantial reason for the decrease in the quality of care in hospitals. As one participant noted,

*"You are losing strong expertise… on the other side, you are receiving weak personnel, which is having a major effect on the quality of care… you find someone who graduated only two years ago, and he is working in critical care… he needs a lot of time to perform well…" – Key Informant 8.*

Another key informant added:

*"Last time, a patient asked for a neurosurgeon… I was shocked to discover that we have only one in the hospital… imagine, we only have this choice… and he is working in 7 or 8 hospitals…" – Key Informant 4*

The ratio of HCW to patients also changed significantly. According to one nursing supervisor:

*"Before the crisis, the ratio of nurses to patients was 1 to 7… now, it is 1 to 12… so, of course, the quality of care will be impacted…" – Key Informant 11*

In addition to the loss of qualified HCW, the heavy workload has a massive impact on the quality of care provided. Nine respondents reported that HCW are struggling to perform clinical tasks due to excessive workload. Many supervisors have had to take on both technical and management duties to fill these gaps:

*"At many times, there is no one to fill the gap, but me… I must perform technical and management tasks… I am connecting the dots…" – Key Informant 4*

One pharmacist reported:

*"We are not able to provide the same quality of service as before… we stopped educating patients about the medicines when dispensing them… now, I cannot ensure that patients understand everything…" – Key Informant 5*

Additionally, key informants reported being unable to take vacations due to the workload, which negatively affects the quality of their work:

*"As a direct impact… we are not able to take a vacation… we cannot take a break… sometimes, we have to take our work home…" – Key Informant 2*

The current economic situation has forced HCW to work in multiple health institutions to cover basic expenses. This, combined with high migration rates, has left HCW exhausted, drained, and unproductive. As one supervisor mentioned,

*"There is a heavy load on nurses working in other hospitals… we notice that… many issues have been occurring recently because they are not functioning well… they have night duties in one hospital, and they work with us during the day… I cannot tell them to leave that other work… and at the same time, I must give them duties…" – Key Informant 7*

The quality of care was also affected by the closure of some departments. Six interviewees mentioned that the closure of departments due to high staff migration rates and COVID-19 situations has impacted the quality of care:

*"At some point, we had to close the chemotherapy floor because we did not have enough registered nurses… only experienced registered nurses can give chemotherapy… so, we had to close the floor because you cannot risk the life of the patient…" – Key Informant 10*

The migration of HCW has also impacted development programs at the hospital level. Four interviewees affirmed that the shortage of staff hampers the ability to conduct continuous training and development workshops:

*"Usually, we need from 6 months to 2 years to train HCW… now, we have to train them in two weeks because of staff shortages…" – Key Informant 1*

4. Retention strategies

Financial strategies were repeatedly highlighted by participants as crucial for retaining HCW, particularly in the context of the ongoing economic crisis. Many key informants noted that various financial incentives had been implemented, including opening additional health centers funded by private and international organizations, providing extra payments for staff working on the COVID-19 floor, and offering monthly social assistance. One participant explained,

> "They worked in parallel with the XXX center… there they give more money… for example, when I work my 12-hour duty for 1 million Lebanese pounds… it's different than working for 150 thousand… this motivates the staff." – Key Informant 7

Despite these efforts, participants emphasized that the devaluation of the Lebanese pound had rendered such payments insufficient, with one informant noting

> "Every month, they tell us we will get social assistance… it's true, I now have double the salary I used to have… but my salary was $1500 to $2000… today, with the Lebanese pound devaluation, I'm earning 7 million, but it's worthless compared to dollars." – Key Informant 7

Several participants suggested paying part of salaries in US dollars and increasing basic salaries to improve retirement benefits,

> "They should give part of the salary in fresh dollars… they can do it as an administrative action… some public hospitals are already doing this." – Key informant 11

while others recommended additional compensation for extra shifts to better motivate staff. Attempts to supplement income through part-time jobs, however, were described as counterproductive, often leaving employees exhausted and reducing overall productivity.

> "The administration tried to find extra jobs for the staff… but this paralyzes the hospital… employees come to work exhausted from their other jobs." – Key Informant 8

Creating a supportive working environment was identified as another key retention strategy. Participants emphasized that appreciation, emotional support, constant communication, effective leadership, and social activities contributed to staff motivation and wellbeing. Sending thank-you messages for extra efforts, fostering open dialogue about personal concerns, maintaining online groups and regular meetings, and involving staff in decision-making were all mentioned as ways to strengthen morale and encourage commitment. One participant noted,

> "Being a good leader encourages staff to work more productively… involving staff in decision-making and giving them responsibility roles boosts morale" – Key Informant 9.

Others highlighted the role of social activities, such as access to recreational facilities, in enhancing psychological wellbeing.

Regulatory strategies at the hospital level were also reported as important for minimizing staff turnover. Flexible work schedules, clearly defined recruitment criteria, prioritizing long-term staff, rotation systems, permanent contracts, and management of unpaid leaves were all described as mechanisms to retain personnel while maintaining operational continuity. For example, participants explained how rotation systems allow staff to cover multiple sections and prevent service gaps when colleagues leave, while permanent contracts provide job security and stability.

Career development emerged as another critical component of retention. Several key informants stressed the importance of continuous training and advancement opportunities, arguing that financial incentives alone are insufficient if staff do not perceive clear pathways for professional growth. As one department supervisor stated,

*"The economic situation is very important… but what comes after it is the chance for advancements… in posts and positions… staff need to feel that there are progression plans… they need to see opportunities for development and higher positions to aim for." – Key Informant 3*

Finally, sustainability was emphasized as a fundamental aspect of any retention strategy. While financial strategies were seen as immediately impactful, participants cautioned that their effectiveness depends on consistent implementation. One informant summarized, *"There is nothing sustainable unless they pay in fresh dollars" – Key Informant 12.*

## Discussion

This mixed-methods study examined migration intentions among HCWs at RHUH amid Lebanon's overlapping economic, political, and social crises. The findings indicate a high prevalence of migration intentions among HCWs, largely driven by financial dissatisfaction and concerns about security. Qualitative data corroborated these results, documenting substantial workforce attrition since 2019, which has adversely affected service delivery through staff shortages, increased workloads, and occasional department closures. Key informants emphasized the importance of financial incentives, professional growth opportunities, supportive management, and flexible work arrangements as essential measures to strengthen workforce retention.

Combining findings from the quantitative and qualitative data, our findings underscore a critical situation, with 70.0% of surveyed HCW expressing intentions to migrate. In addition, key informant interviews further elucidated the widespread nature of migration since 2019, with many HCW opting for destinations in Gulf countries and Europe, where economic opportunities and security are perceived to be more favorable. This high level of migration intentions among Lebanese HCW reflects a broader trend of skilled professionals seeking opportunities abroad, driven by economic hardship and insecurity at home. The economic crisis in Lebanon, marked by the rapid devaluation of the local currency and inflation, has drastically reduced the purchasing power of HCW' salaries paid in Lebanese pounds [15]. This situation poses a severe threat to HCW' financial security and undermines efforts to retain them within the healthcare sector. As a result, many feel compelled to seek better remuneration and stability elsewhere.

The findings of the present study align with previous research on the push and pull factors driving HCW migration from Lebanon [14,20,22,25,33,34]. In particular, our results are consistent with a prior study focusing on Lebanese nurses [22], which also identified unsatisfactory benefits or salaries as the primary driver of migration. In our quantitative analysis, 91% of participants reported financial concerns as a key factor, compared to 72.8% in the study by Alameddine et al. [22], underscoring the persistent economic challenges within Lebanon's healthcare sector. Furthermore, our regression analysis demonstrated that lower satisfaction with pay significantly increases the likelihood of intending to migrate, highlighting the critical role of financial remuneration in healthcare workers' migration decisions.

The current study underscores a notable increase in security concerns cited by 81% of respondents in the current study, a significantly greater proportion than the 35% reported in earlier studies [22]. This stark difference highlights an intensified perception of insecurity among HCW amidst Lebanon's ongoing economic crisis and political instability. This is supported by the literature that cites political instability in Lebanon and the region as a major reason for the migration of healthcare professionals [14,18,35]. Historical events, such as the Lebanese civil war in the 1970s, were significant factors driving the migration of physicians [34]. Similarly, civil wars in countries such as Syria, Iraq, and Libya have led to the migration of health professionals due to concerns about training, security, and finances [36]. Other developing countries, such as Mozambique and Liberia in Sub-Saharan Africa, have also experienced substantial nursing

workforce migration due to civil wars, with reports indicating that up to 80% of their needed nursing staff migrated abroad [37]. Providing a secure and stable environment for Lebanese nurses could be a crucial strategy to attract them back to the country.

The quantitative analysis also identified the lack of a supportive environment as a reason for migration intention, a factor not mentioned in the previous study [22]. This highlights an evolving understanding of the challenges faced by HCW in Lebanon. Emphasizing a supportive environment underscores the importance of workplace culture and morale, which are crucial for job satisfaction and retention. Effective management, collegial relationships, professional recognition, and emotional support are vital [38]; without these factors, HCW may feel undervalued and isolated, prompting them to seek employment in more supportive settings abroad. Moreover, the current study provides deeper insights into additional personal motivations such as family relocation and career diversification outside traditional healthcare settings.

The findings from the key informants highlight the severe impact of HCW migration on Lebanese hospitals. The loss of experienced professionals is leading to a decline in the quality of care, with hospitals relying on less experienced staff who need more time to develop critical skills. This mismatch between patient needs and staff capabilities compromises patient outcomes. Additionally, heavier workloads on remaining staff and the closure of departments due to staff shortages further undermine patient safety and service quality. The increased burden on remaining HCW also leads to significant burnout, exacerbating the challenges faced by the healthcare system and further jeopardizing patient care. These observations align with findings from previous studies, which demonstrate that higher patient-to-nurse ratios adversely affect healthcare outcomes. One study found that increased ratios in Pennsylvania hospitals led to higher 30-day patient mortality and failure-to-rescue rates, as well as a significant rise in nurse burnout and job dissatisfaction [37]. Together, these findings suggest that higher HCW workloads and lower educational qualifications can negatively impact patient outcomes and HCW well-being, highlighting the risks of cost-cutting measures in HCW staffing.

Combining insights from HCW and key informants underscores the urgent need for coordinated efforts from policymakers and healthcare leaders to implement targeted strategies aimed at retaining and attracting HCW in Lebanon. While factors such as economic and political instability pose challenges beyond healthcare organizations' control, our findings highlight actionable strategies to mitigate HCW' migration. Enhancing financial incentives, such as adjusting pay scales to counter inflation and introducing partial salary payments in stable currencies, is crucial for alleviating economic pressures that drive migration. Key Informant 12's perspective underscores the sustainability challenge of retention strategies amid Lebanon's volatile economic climate, emphasizing the importance of stable income practices. Improving working conditions by fostering a supportive environment, reducing workloads, implementing flexible schedules, ensuring fair recruitment, offering permanent contracts, and investing in career development are essential for enhancing job satisfaction and retaining HCW. Additionally, fostering international collaboration can significantly enhance Lebanon's healthcare capacity amidst ongoing crises by facilitating knowledge transfer and strengthening emergency response, preparedness, capacity building, and healthcare infrastructure.

## Strengths and limitations

This study has several strengths that contribute to its significance. Unlike prior studies that often concentrate solely on nurses or medical students [20–22], this research encompasses a broader spectrum, including physicians, allied health practitioners, technicians, and administrative staff. By examining migration drivers across these diverse categories, this study provides a comprehensive understanding of Lebanon's healthcare workforce dynamics amidst economic and political challenges. These insights are essential for crafting targeted strategies to mitigate the impact of HCW migration on healthcare systems globally, ensuring sustained and high-quality healthcare delivery despite workforce mobility. The mixed-methods approach, which combines quantitative surveys with qualitative interviews, offers a robust and nuanced understanding of the issue, capturing both statistical trends and personal experiences. The use of purposive sampling to select department chiefs ensures that the perspectives of key decision-makers are included, adding depth to the findings.

Several limitations should be considered when interpreting the findings of this study on HCW' migration intentions in Lebanon. First, the sample size of 120 participants from a single hospital limits the generalizability of the results to the broader healthcare workforce across Lebanon. Moreover, the reliance on self-reported data introduces the possibility of recall bias and subjective interpretations of survey questions. The qualitative nature of interviews also entails inherent biases in data interpretation, including potential bias related to the positions and roles of the interviewees, as department chiefs may have perspectives that differ from other staff members, which could influence the themes identified despite efforts to maintain rigor through systematic analysis. Additionally, the study captures a specific period of Lebanon's socio-economic and political turmoil, limiting the temporal validity and applicability of the findings to other contexts or different phases of stability. Despite these limitations, this study provides critical insights into the factors influencing HCW' migration intentions amid Lebanon's socioeconomic turmoil.

## Conclusion

In conclusion, our findings underscore the evolving dynamics and heightened challenges influencing healthcare workforce migration in Lebanon. Addressing economic issues and improving overall working conditions are critical to mitigating the migration intentions of HCW. Enhancing financial incentives and adjusting pay scales to counter inflation and considering partial payments in stable currencies can alleviate economic pressures. Improving working conditions by fostering a supportive and safe environment, reducing workloads, and ensuring effective management can significantly boost job satisfaction. Additionally, fostering international collaboration for knowledge transfer, resource sharing, and joint healthcare initiatives can enhance the resilience of Lebanon's healthcare system amidst ongoing economic and political challenges. By implementing these comprehensive strategies, Lebanon can better retain its healthcare workforce, ensuring sustainable and high-quality healthcare delivery for its population.

## Supporting information

**S1 Appendix. Interview topic guide.**
(DOCX)

**S1 Table. Summary of staff suggestions for improving retention at RHUH.**
(DOCX)

## Acknowledgments

The authors would like to thank all the healthcare workers at RHUH who accepted to participate in the study for their time and cooperation.

## Author contributions

**Conceptualization:** Maya Hassan, Elizabeth Al Mounajed, Hala Dalal.

**Data curation:** Maya Hassan, Elizabeth Al Mounajed.

**Formal analysis:** Linda Abou-Abbas, Maya Hassan.

**Methodology:** Maya Hassan, Hala Dalal.

**Project administration:** Maya Hassan.

**Supervision:** Elizabeth Al Mounajed, Hala Dalal.

**Writing – original draft:** Linda Abou-Abbas, Maya Hassan.

**Writing – review & editing:** Elizabeth Al Mounajed, Mohamad Shafic Ramadan, Hala Dalal.

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
