## [Decision Letter · Decision Letter 0]

22 Dec 2025

PONE-D-25-55984Migration Intentions of Healthcare Workers in a Lebanese Public University Hospital Amid Crises: A Mixed Method Study PLOS One

Dear Dr. Abou Abbas,

Thank you for submitting your manuscript to PLOS ONE. After careful consideration, we feel that it has merit but does not fully meet PLOS ONE’s publication criteria as it currently stands. Therefore, we invite you to submit a revised version of the manuscript that addresses the points raised during the review process. Please submit your revised manuscript by Feb 05 2026 11:59PM. If you will need more time than this to complete your revisions, please reply to this message or contact the journal office at plosone@plos.org . Please include the following items when submitting your revised manuscript:

We look forward to receiving your revised manuscript.

Kind regards,

Majed Sulaiman Alamri, PhD

Academic Editor

PLOS One

Journal Requirements:

3. In the online submission form you indicate that your data is not available for proprietary reasons and have provided a contact point for accessing this data. Please note that your current contact point is a co-author on this manuscript. According to our Data Policy, the contact point must not be an author on the manuscript and must be an institutional contact, ideally not an individual. Please revise your data statement to a non-author institutional point of contact, such as a data access or ethics committee, and send this to us via return email. Please also include contact information for the third party organization, and please include the full citation of where the data can be found.

5. Please amend your list of authors on the manuscript to ensure that each author is linked to an affiliation. Authors’ affiliations should reflect the institution where the work was done (if authors moved subsequently, you can also list the new affiliation stating “current affiliation:….” as necessary).

6. Please amend either the abstract on the online submission form (via Edit Submission) or the abstract in the manuscript so that they are identical.

Additional Editor Comments:

Please address reviewers’ comments.

Reviewers' comments:

Reviewer's Responses to Questions

**Comments to the Author**

1. Is the manuscript technically sound, and do the data support the conclusions?

Reviewer #1: Yes

Reviewer #2: Yes

2. Has the statistical analysis been performed appropriately and rigorously? 

Reviewer #1: Yes

Reviewer #2: Yes

3. Have the authors made all data underlying the findings in their manuscript fully available?

Reviewer #1: Yes

Reviewer #2: Yes

4. Is the manuscript presented in an intelligible fashion and written in standard English?

Reviewer #1: Yes

Reviewer #2: Yes

5. Review Comments to the Author

Reviewer #1: Dear Authors, this is a methodologically rigorous study and a well written manuscript. There are a few edits that are needed. For example, on line 100 it states, "Lebanese lira" on line 402 it says, "Lebanese pound." There are also inconsistencies in tense. For example, on line 109, "Studies in Lebanon have highlighted..."

I look forward to reading the published article.

Reviewer #2: General comments:

This is a good study with appropriate details given and good flow of narrative.

Abstract:

1) Perhaps want to include the odds ratio for multiple regression result.

2) Could probably cut the proportion of women as majority as unsure how this may change the narrative

3) Reading the whole thing, I would say that the qualitative part focuses a lot about impact of migration, perhaps want to revise title to include that word. Maybe readers will have impression that this tells two points of views from the workers (quanti) and upper management (quali)

Introduction:

4) Page 4. Simultaneously, systemic issues such as weak health systems � what does weak systems mean)

5) Page 5: Since October 2019…. � maybe could show better context of what’s been happening over the last few years with specific year of things happening – unsure explosion happened when, so might be better for new readers to understand when each occasion happened.

6) Page 5: Lebanese lira � could put in bracket (Lebanese currency)

7) Page 6: say that Beirut is capital of Lebanon, maybe also could put the population number so that readers could get a sense how big is the capital and how big is the demand

8) Maybe explain a bit about what you mean by political instability in Lebanon, quite vague and general

9) Last paragraph: Make a distinction of doing quanti study to HCW and quali to management levels

Materials and methods:

10) Can you say why the medical residents and department chiefs were excluded. If residents mean in training, does that mean no other trainees that were excluded like nurses, etc

11) Quali phase Inclusion: …. Willing to participate (I don’t think this sentence is necessary as this is why you include them as eligible. Consent part comes after that). In that sense, does that mean the exclusion criteria are those who participated in the quantitative survey, perhaps can write that so it makes a distinction of HCWs who answered their reasons to migrate (bottom up approach) and management POVs why this is happening (top down approach)

12) Out of curiosity, are all the HCWs have Lebanese nationalities, meaning no international workers so far?

13) Noted in introduction that migration was reported among nurses or medical doctor. As you are also including pharmacists, dietitition, etc,..is there evidence for this as well that should be the intro, or you are purposely do this as a part of seeing whats the trend in other HCWs that were not usually reported as well

Sampling strategy

14) 550 HCW – is this the total number of all the staffs in the hospital and where is this number obtained from; registrar, admin? Also saying that this is the target population for the quantitative part or overall?

15) Stratified sampling – understand the logic, but whats the number like in terms of distributions across specialties

16) Purposive sampling – what does this mean, and also would help if there are flowchart of sample selection from eligible, excluded, and final

17) For quali sample, does that mean the people selected were ensured from different department

Data collection tools

18) Work experience - …..salary range.. ( is this a monthly based salary)

19) Page 10 (HCW migration intentions) – the last sentence implied that they were asked about the reasons and potential strategies. Does this mean sort of open ended question in the survey. I can see the result from Figure 2, but these were not shown in the method part

Data analysis

20) Page 12 – first two sentences were repeated

21) Unsure if the data categorisations in the survey is similar to the analyses. Is there any recategorization due to smaller distribution in subcategory. Or data transformation after seeing the total numbers.

22) Translation was reviewed…… - is this done by consensus of the authors or had third part doing it

Results

23) Response rates – give a percentage, also is there a reason they did not participate (was it accessed in the survey if HCW were not consented). Also, could be more explicit that the first part is the quantitative analysis as this 120 people were the sample and 10 is for qualitative

24) Table 1: ensure if want to use one decimal point for proportions then all are shown one decimal point (some are missing). Also, educational background BT/LT (what does this acronym mean). Salaray range is not sure if this implies an hourly rate, in lebanese currency?

25) Page 15 – Of the total respondents (79.2%)…. This implies that from the 79.2%, 20.8% had another job. Don’t think that is quite right. The 79.2% is not needed in the text

26) Bivariate analysies examining……Additionally, our analysis….. several things confuses me in this part. The numbers in the texts are not the same as the table. Also, the text didn’t say what is dropped or included in the multiple logistic regression analyses. Table shows only adjusted OR for pay (which numbers are not the same), but text shows it includes prospects as well, and the decimal points were not similar presented. If all wants to shows three decimal points for 95%CI, please keep all the same (or two decimal points)

27) Table 3. I don’t think p-value adds any value or needed here, as simple logistic regression shows the similar bivariate association. (less reliance to p-values but more focus on the estimates)

28) Staff suggestions – had troubled understanding where this analyses came from. Refer to method, where this should be explicitly said what is the sentence in the survey that leads to this sections. Is it write whatever or guided like please write about financial, or support, management treatment, etc

29) Qualitative part: Findings from key informant… - years of experience does this imply working in the hospital, or in general, or in Lebanon

30) Page 19: "The dollar rate is increasing... and we are still getting our salary on the 1,500 rates... – what does the meaning of rates.

31) Two key informants mentioned the possibility of obtaining another 440 passport, which offers greater mobility and legal security….this sentence repeated twice in the same paragraph

32) Page 22: The quality of care is also affected…. – change to past tense. Also RN abbreviation needs the full name

33) Page 24: Flexible work schedules, thoughtful recruitment criteria,… - what is the meaning of ‘thoughtful recruitment criteria’

Discussion

34) Unlike prior studies that often concentrate solely on nurses or medical students [20-22], this research encompasses a broader spectrum, including physicians, allied 572 health practitioners, technicians, and administrative staff. By examining migration drivers across these diverse categories, this study provides a comprehensive understanding of Lebanon's healthcare workforce dynamics amidst economic and political challenges. These insights are essential for crafting targeted strategies to mitigate the impact of HCW migration on healthcare systems globally, ensuring sustained and high-quality healthcare delivery 577 despite workforce mobility. I feel like this is one of the strengths of this study, perhaps rearrange this to put it there? Cause the first paragraph should refer to the main aim, and I don’t think is the main point that should be raised.

35) Combining findings from the quantitative and qualitative data… - perhaps this fit much better as the first paragraph in discussion

36) Page 28: The quantitative analysis also identified the lack of a supportive environment as a reason for migration intention, a factor not mentioned inthe previous study.—which previous study. Same paragraph for this part: Effective management, collegial relationships, professional recognition, and emotional support are vital; without these factors, HCW may feel undervalued and isolated, prompting them to seek employment in more supportive settings abroad (any reference)

37) Page 29: ….higher proportion of nurses with bachelor's degrees decreased this likelihood. I don’t think this necessary as this does not analyse in your study and a different findings from your results as well

References:

Please refer to guidelines for all (no bold, no Italic format) and revise formatting for website sources in references 1 and 26

6. PLOS authors have the option to publish the peer review history of their article (what does this mean? ). If published, this will include your full peer review and any attached files.

**Do you want your identity to be public for this peer review?** For information about this choice, including consent withdrawal, please see our Privacy Policy .

Reviewer #1: **Yes:** Magdeline Aagard, EdD, MBA, BAN

Reviewer #2: No

---

## [Author Response · Author response to Decision Letter 1]

29 Dec 2025

December 22, 2025

Dear Dr. AlAmri,

We are pleased to resubmit the revised version of Manuscript entitled: “Migration Intentions and their impact on Healthcare Workers in a Lebanese Public University Hospital Amid Crises: A Mixed Method Study ”

Ref: Submission ID: PONE-D-25-55984

We would like to thank the editor and the reviewers for their constructive and insightful comments. We have carefully addressed all comments. All responses to the reviewers’ comments are provided in italics and prefaced by “Author response.” Modifications in the manuscript are highlighted in yellow. Please find below a summary of how we addressed the comments.

My co-authors and I sincerely appreciate the opportunity to resubmit and are enthusiastic about the possibility of publication.

Reviewer #1: Dear Authors, this is a methodologically rigorous study and a well written manuscript. There are a few edits that are needed. For example, on line 100 it states, "Lebanese lira" on line 402 it says, "Lebanese pound." There are also inconsistencies in tense. For example, on line 109, "Studies in Lebanon have highlighted..."

I look forward to reading the published article.

Author’s reply: We sincerely thank the reviewer for their positive feedback and careful review. We have standardized all references to the currency throughout the manuscript and corrected inconsistencies in tense to ensure clarity and consistency. We appreciate your constructive comments, which have improved the manuscript.

Reviewer #2: General comments:

This is a good study with appropriate details given and good flow of narrative.

Abstract:

1) Perhaps want to include the odds ratio for multiple regression result.

Author’s reply: Added

2) Could probably cut the proportion of women as majority as unsure how this may change the narrative

Author’s reply: Thank you for your suggestion. We have revised the sentence to focus on migration intentions and the main drivers, while de-emphasizing gender, as follows: Of 120 healthcare workers surveyed, 70% intended to migrate, mainly due to financial (93%) and security concerns (81%).

3) Reading the whole thing, I would say that the qualitative part focuses a lot about impact of migration, perhaps want to revise title to include that word. Maybe readers will have impression that this tells two points of views from the workers (quanti) and upper management (quali).

Author’s reply: Thank you for your insightful comment. We agree that the qualitative section emphasizes the impact of migration. The title has been revised to better reflect this focus:

Title: Migration Intentions and their impact on Healthcare Workers in a Lebanese Public University Hospital Amid Crises: A Mixed Method Study

Introduction:

4) Page 4. Simultaneously, systemic issues such as weak health systems � what does weak systems mean)

Author’s reply: We thank the reviewer for highlighting this point. To improve clarity, we have revised the wording to better define what is meant by “weak health systems.” Specifically, we now describe these systems as under-resourced and poorly organized.

5) Page 5: Since October 2019…. � maybe could show better context of what’s been happening over the last few years with specific year of things happening – unsure explosion happened when, so might be better for new readers to understand when each occasion happened.

Author’s reply: We thank the reviewer for this helpful suggestion. To provide clearer chronological context for readers, we have revised the text to explicitly specify the timing of the major crises affecting Lebanon, including the onset of the economic crisis (2019), the COVID-19 pandemic (2020–2021), and the Beirut port explosion (August 2020).

6) Page 5: Lebanese lira � could put in bracket (Lebanese currency)

Author’s reply: Corrected.

7) Page 6: say that Beirut is capital of Lebanon, maybe also could put the population number so that readers could get a sense how big is the capital and how big is the demand.

Author’s reply: We thank the reviewer for this valuable suggestion. We have revised the manuscript to clarify that Beirut is the capital of Lebanon and added population context to help readers better understand the scale of healthcare demand served by Rafik Hariri University Hospital.

8) Maybe explain a bit about what you mean by political instability in Lebanon, quite vague and general

Author’s reply: We thank the reviewer for this comment. To address this concern, we have clarified what is meant by political instability in the Lebanese context by briefly describing prolonged governmental paralysis, recurrent political unrest, and weak institutional governance, all of which have affected public services, economic stability, and the healthcare sector. This clarification has been added to improve precision and contextual understanding for readers.

9) Last paragraph: Make a distinction of doing quanti study to HCW and quali to management levels

Author’s reply: We have revised the manuscript to clarify the study design, specifying that the quantitative component targeted frontline healthcare workers to assess their migration intentions and primary drivers, while the qualitative component focused on management-level staff to explore systemic and organizational factors influencing workforce migration.

Materials and methods:

10) Can you say why the medical residents and department chiefs were excluded. If residents mean in training, does that mean no other trainees that were excluded like nurses, etc

Author’s reply: Medical residents were excluded because they are still in training and may have different career trajectories, work expectations, and migration considerations compared to fully licensed healthcare professionals. Similarly, department chiefs were excluded as their perspectives reflect managerial responsibilities rather than frontline clinical experiences, which may bias responses regarding migration intentions and drivers. Other trainee or support staff, such as nurses in formal training programs, were not included in the exclusion criteria because the study focused on HCWs employed in regular, active clinical or operational roles at RHUH, ensuring that the findings reflect the migration intentions of staff with permanent or contractual responsibilities.

11) Quali phase Inclusion: …. Willing to participate (I don’t think this sentence is necessary as this is why you include them as eligible. Consent part comes after that). In that sense, does that mean the exclusion criteria are those who participated in the quantitative survey, perhaps can write that so it makes a distinction of HCWs who answered their reasons to migrate (bottom up approach) and management POVs why this is happening (top down approach)

Author’s reply: We thank the reviewer for this comment. Clarified in the methods section

12) Out of curiosity, are all the HCWs have Lebanese nationalities, meaning no international workers so far?

Author’s reply: Information on participants’ nationality was not collected; therefore, we cannot confirm whether all healthcare workers were Lebanese or if any were international staff.

13) Noted in introduction that migration was reported among nurses or medical doctor. As you are also including pharmacists, dietitition, etc,..is there evidence for this as well that should be the intro, or you are purposely do this as a part of seeing whats the trend in other HCWs that were not usually reported as well

Author’s reply: Thank you for this comment. The existing literature primarily reports migration among nurses and medical doctors. We included pharmacists, dietitians, and other healthcare workers to explore whether similar migration trends are present among these professional categories, which are less frequently studied. This allows us to provide a broader picture of workforce intentions and potential gaps across all staff categories in the hospital.

Sampling strategy

14) 580 HCW – is this the total number of all the staffs in the hospital and where is this number obtained from, registrar, admin? Also saying that this is the target population for the quantitative part or overall?

Author’s reply: Thank you for your comment. We confirm that the target population of approximately 580 healthcare workers represents all eligible staff employed at RHUH at the time of the study, as obtained from the hospital’s human resources and administrative registries. This number specifically applies to the quantitative phase of the study, and a stratified sampling approach was used to ensure proportional representation across all professional categories.

15) Stratified sampling – understand the logic, but what’s the number like in terms of distributions across specialties

Author’s reply: Thank you for your comment. The target population of HCWs at RHUH comprised 579 staff members across various professional categories. Stratified sampling was used to ensure proportional representation across all specialties. The distribution of staff was as follows: nurses (n = 400), attending doctors (n = 100), laboratory technicians and phlebotomists (n = 30), pharmacists (n = 12), inhalation specialists (n = 10), physiotherapists (n = 5), and dietitians (n = 2).

16) Purposive sampling – what does this mean, and also would help if there are flowchart of sample selection from eligible, excluded, and final

Author’s reply: We thank the reviewer for the comment. Purposive sampling is a non-probability sampling technique in which participants are selected based on specific characteristics relevant to the study objectives. In our study, purposive sampling was used for the qualitative phase to select key informants (department chiefs and managerial staff) who could provide in-depth insights into the drivers of healthcare worker migration and systemic factors at RHUH.

To enhance clarity, we have also prepared a flowchart showing the sample selection process, including the total eligible population, exclusions, and the final number of participants included in both the quantitative and qualitative phases. This flowchart was added to the revised manuscript to visually summarize the sampling strategy.

17) For quali sample, does that mean the people selected were ensured from different department

Author’s reply: yes

Data collection tools

18) Work experience - …..salary range.. ( is this a monthly based salary)

Author’s reply: Yes Corrected.

19) Page 10 (HCW migration intentions) – the last sentence implied that they were asked about the reasons and potential strategies. Does this mean sort of open ended question in the survey. I can see the result from Figure 2, but these were not shown in the method part

Author’s reply: We have clarified the Methods section to explicitly state that reasons for migration were collected using structured, predefined response options, while potential retention strategies were explored through an open-ended question. The relevant text in the methods section has been revised to improve clarity.

Data analysis

20) Page 12 – first two sentences were repeated

Author’s reply: Thank you for pointing this issue. Corrected.

21) Unsure if the data categorisations in the survey is similar to the analyses. Is there any recategorization due to smaller distribution in subcategory. Or data transformation after seeing the total numbers.

Author’s reply: During analysis, some subcategories with very small counts were combined into broader categories to ensure meaningful interpretation and maintain statistical robustness. No additional data transformations were performed beyond these recategorizations.

22) Translation was reviewed…… - is this done by consensus of the authors or had third part doing it

Author’s reply: The translations were performed by one member of the research team and subsequently reviewed by another team member for accuracy and consistency with the original transcripts. Any discrepancies were resolved through discussion within the research team.

Results

23) Response rates – give a percentage, also is there a reason they did not participate (was it accessed in the survey if HCW were not consented). Also, could be more explicit that the first part is the quantitative analysis as this 120 people were the sample and 10 is for qualitative.

Author’s reply: Thank you for your comments. We have clarified the sample and response rates in the revised manuscript.

24) Table 1: ensure if want to use one decimal point for proportions then all are shown one decimal point (some are missing).

Author’s reply: Corrected

Also, educational background BT/LT (what does this acronym mean).

Author’s reply: Corrected

Salaray range is not sure if this implies an hourly rate, in lebanese currency?

Author’s reply: Monthly Salary Range. Corrected

25) Page 15 – Of the total respondents (79.2%)…. This implies that from the 79.2%, 20.8% had another job. Don’t think that is quite right. The 79.2% is not needed in the text

Author’s reply: Thank you for pointing this issue. Corrected.

26) Bivariate analysies examining……Additionally, our analysis….. several things confuses me in this part. The numbers in the texts are not the same as the table. Also, the text didn’t say what is dropped or included in the multiple logistic regression analyses. Table shows only adjusted OR for pay (which numbers are not the same), but text shows it includes prospects as well, and the decimal points were not similar presented. If all wants to shows three decimal points for 95%CI, please keep all the same (or two decimal points).

Author’s reply: Corrected

27) Table 3. I don’t think p-value adds any value or needed here, as simple logistic regression shows the similar bivariate association. (less reliance to p-values but more focus on the estimates)

Author’s reply: The column of the P-value was deleted.

28) Staff suggestions – had troubled understanding where this analyses came from. Refer to method, where this should be explicitly said what is the sentence in the survey that leads to this sections. Is it write whatever or guided like please write about financial, or support, management treatment, etc

Author’s reply: We have clarified this in the Methods section by specifying that staff were asked an open-ended question. In the Results section, we have added one sentence explicitly stating that this analysis is based on responses to that open-ended question.

29) Qualitative part: Findings from key informant… - years of experience does this imply working in the hospital, or in general, or in Lebanon

Author’s reply: in general. Corrected.

30) Page 19: "The dollar rate is increasing... and we are still getting our salary on the 1,500 rates... – what does the meaning of rates.

Author’s reply: We have clarified this by adding a brief explanatory note in the text to indicate that “rates” refers to the exchange rate at which salaries are calculated.

31) Two key informants mentioned the possibility of obtaining another 440 passport, which offers greater mobility and legal security…. this sentence repeated twice in the same paragraph

Author’s reply: corrected

32) Page 22: The quality of care is also affected…. – change to past tense. Also RN abbreviation needs the full name

Author’s reply: Corrected

33) Page 24: Flexible work schedules, thoughtful recruitment criteria,… - what is the meaning of ‘thoughtful recruitment criteria’.

Author’s reply: We have revised this phrase to improve clarity. “Thoughtful recruitment criteria” refers to “clearly defined recruitment criteria”. The wording has been revised accordingly in the manuscript.

Discussion

34) Unlike prior studies that often concentrate solely on nurses or medical students [20-22], this research encompasses a broader spectrum, including physicians, allied 572 health practitioners, technicians, and administrative staff. By examining migration drivers across these diverse categories, this study provides a comprehensive understanding of Lebanon's healthcare workforce dynamics amidst economic and political challenges. These insights are essential for crafting targeted strategies to mitigate the impact of HCW migration on healthcare systems globally, ensuring sustained and high-quality healthcare delivery 577 despite workfor

---

## [Decision Letter · Decision Letter 1]

7 Jan 2026

Migration Intentions and their impact on Healthcare Workers in a Lebanese Public University Hospital Amid Crises: A Mixed Method Study

PONE-D-25-55984R1

Dear Dr. Linda,

We’re pleased to inform you that your manuscript has been judged scientifically suitable for publication and will be formally accepted for publication once it meets all outstanding technical requirements.

Kind regards,

Majed Sulaiman Alamri, PhD

Academic Editor

PLOS One

Additional Editor Comments (optional):

Reviewers' comments:

Reviewer's Responses to Questions

**Comments to the Author**

1. If the authors have adequately addressed your comments raised in a previous round of review and you feel that this manuscript is now acceptable for publication, you may indicate that here to bypass the “Comments to the Author” section, enter your conflict of interest statement in the “Confidential to Editor” section, and submit your "Accept" recommendation.

Reviewer #1: All comments have been addressed

Reviewer #2: All comments have been addressed

2. Is the manuscript technically sound, and do the data support the conclusions?

Reviewer #1: Yes

Reviewer #2: Yes

3. Has the statistical analysis been performed appropriately and rigorously? 

Reviewer #1: Yes

Reviewer #2: Yes

4. Have the authors made all data underlying the findings in their manuscript fully available?

Reviewer #1: Yes

Reviewer #2: Yes

5. Is the manuscript presented in an intelligible fashion and written in standard English?

Reviewer #1: Yes

Reviewer #2: Yes

6. Review Comments to the Author

Reviewer #1: Dear Authors, thank you for your thorough review of our comments. You have addressed each one, providing detailed responses and correcting them within the manuscript.

Reviewer #2: Great work ! Thank you for addressing all the comments with good explanations and great elaborations. A very good study that will help more understanding on the migration factors for healthcare works and its impact, especially in Lebanon.

7. PLOS authors have the option to publish the peer review history of their article (what does this mean? ). If published, this will include your full peer review and any attached files.

**Do you want your identity to be public for this peer review?** For information about this choice, including consent withdrawal, please see our Privacy Policy .

Reviewer #1: **Yes:** Magdeline Aagard

Reviewer #2: **Yes:** Abdul Azim Abdul Razak

---

## [Editor Report · Acceptance letter]

PONE-D-25-55984R1

PLOS One

Dear Dr. Abou-Abbas,

I'm pleased to inform you that your manuscript has been deemed suitable for publication in PLOS One. Congratulations! Your manuscript is now being handed over to our production team.

Kind regards,

on behalf of

Prof. Majed Sulaiman Alamri

Academic Editor

PLOS One